# Toll-like receptor 4 and macrophage scavenger receptor 1 crosstalk regulates phagocytosis of a fungal pathogen

Chinaemerem U. Onyishi [1], Guillaume E. Desanti[1], Alex L. Wilkinson [1], Samuel Lara-Reyna [1], Eva-Maria Frickel [1], Gyorgy Fejer[2], Olivier D. Christophe[3], Clare E. Bryant [4], Subhankar Mukhopadhyay [5], Siamon Gordon [6,7] & Robin C. May [1] ✉

The opportunistic fungal pathogen *Cryptococcus neoformans* causes lethal infections in immunocompromised patients. Macrophages are central to the host response to cryptococci; however, it is unclear how *C. neoformans* is recognised and phagocytosed by macrophages. Here we investigate the role of TLR4 in the non-opsonic phagocytosis of *C. neoformans*. We find that loss of TLR4 function unexpectedly increases phagocytosis of non-opsonised cryptococci by murine and human macrophages. The increased phagocytosis observed in *Tlr4*[-/-] cells was dampened by pre-treatment of macrophages with oxidised-LDL, a known ligand of scavenger receptors. The scavenger receptor, macrophage scavenger receptor 1 (MSR1) (also known as SR-A1 or CD204) was upregulated in *Tlr4*[-/-] macrophages. Genetic ablation of MSR1 resulted in a 75% decrease in phagocytosis of non-opsonised cryptococci, strongly suggesting that it is a key non-opsonic receptor for this pathogen. We go on to show that MSR1-mediated uptake likely involves the formation of a multimolecular signalling complex involving FcγR leading to SYK, PI3K, p38 and ERK1/2 activation to drive actin remodelling and phagocytosis. Altogether, our data indicate a hitherto unidentified role for TLR4/MSR1 crosstalk in the non-opsonic phagocytosis of *C. neoformans*.

*Cryptococcus neoformans* is an encapsulated yeast that causes life-threatening infections in humans and other animals[1,2], with an estimated global burden of 181,000 deaths annually[3]. Infection with *C. neoformans* begins with the inhalation of fungal cells from the environment into the lungs[1]. Within the lungs, tissue-resident macrophages are amongst the first immune cells the fungi encounter[4], thus, the interaction between host macrophages and invading fungi is believed to be an important determinant of disease progression and outcome. Non-opsonised cryptococci are phagocytosed poorly[5], but since opsonising antibodies are negligible within the healthy lung[6], this low level of non-opsonic uptake is likely a critical determinant of the subsequent course of an infection. However, there is no clear understanding of the mechanism by which macrophages detect and phagocytose *C. neoformans* in the absence of opsonins[4,7].

[1]Institute of Microbiology & Infection and School of Biosciences, University of Birmingham, Edgbaston, Birmingham B15 2TT, United Kingdom. [2]School of Biomedical Sciences, Faculty of Health, University of Plymouth, Plymouth, United Kingdom. [3]Université Paris-Saclay, INSERM, Hémostase inflammation thrombose HITH U1176, 94276 Le Kremlin-Bicêtre, France. [4]University of Cambridge, Department of Medicine, Box 157, Level 5, Addenbrooke's Hospital, Hills Road, Cambridge CB2 0QQ, United Kingdom. [5]Peter Gorer Department of Immunobiology, School of Immunology & Microbial Sciences, King's College London, London SE1 9RT, United Kingdom. [6]Department of Microbiology and Immunology, College of Medicine, Chang Gung University, Taoyuan, Taiwan. [7]Sir William Dunn School of Pathology, University of Oxford, Oxford, UK. ✉e-mail: r.c.may@bham.ac.uk

Phagocytosis, defined as the uptake of particles greater than 0.5 μm, is a significant process in the innate immune response as it leads to the degradation of invading pathogens and the presentation of microbial ligands on MHC molecules, thereby activating the adaptive arm of the immune system[8]. Non-opsonic phagocytosis is initiated by the recognition of pathogen associated molecular patterns (PAMPs) on the surface of microbes by host pattern recognition receptors (PRRs)[8]. PRRs on professional phagocytes include members of the Toll-like receptor (TLR) family, the C-type lectin receptor (CLR) family, and the scavenger receptor (SR) family. All of these have been implicated in the recognition of *C. neoformans* to varying degrees, with β−1,3-glucans, mannans and glucuronoxylomannan (GXM) found on the *C. neoformans* cell wall and capsule serving as PAMPs[9–15]. The CLR, Dectin-1 (also known as CLEC7A), is well-known for its role in the recognition of fungal β-glucans[11] and has been identified as the key PRR involved in the phagocytosis of *Candida albicans*[16,17]. However, previous work found that Dectin-1 is only marginally involved in the phagocytosis of non-opsonised *C. neoformans*[5], suggesting that other non-opsonic receptors for *C. neoformans* may be more important.

Within the TLR family, TLR4 is known to recognise fungal mannans[12] and GXM[9], leading to the activation of downstream signalling cascades. TLR4 signalling is mediated by the adaptor proteins myeloid differentiation primary response 88 (MyD88) and TIR-domain-containing adapter-inducing interferon-β (TRIF)[18]. The MyD88-dependent pathway is used by all TLRs except TLR3, which uses TRIF-dependent signalling instead[19,20]. The MyD88-dependent pathway and the TRIF-dependent pathway ultimately lead to the activation of the transcription factor nuclear factor κB (NF-κB) and mitogen activated protein kinases (MAPKs)[20]. The TRIF pathway also leads to the activation of Interferon regulatory factor 3 (IRF3). These then act to activate the expression and secretion of proinflammatory cytokines (MyD88 and TRIF pathway) and Type I interferons (TRIF pathway)[8,18,21]. Notably, plasma membrane TLRs also activate Rap GTPase and Rac GTPase to activate phagocytic integrins and other bona fide phagocytic receptors which are then responsible for pathogen engulfment[22].

Whilst investigating the role of TLR signalling in the inflammatory response to cryptococci, we made the unexpected discovery that loss of TLR4 activity leads to enhanced non-opsonic uptake of the fungus. We show that this increase in uptake was driven by crosstalk between TLR4 and Macrophage Scavenger Receptor 1 (MSR1) (also known as SR-A1 or CD204), such that the loss of TLR4 signalling led to elevated cell surface expression of MSR1, but not other SRs, leading to increased uptake. We provide evidence that MSR1 is an important receptor for the non-opsonic phagocytosis of *C. neoformans*, shedding light on a key host receptor involved in the uptake of this fungal pathogen.

## Results

### Both chemical inhibition and genetic loss of TLR4 signalling result in an increase in the phagocytosis of non-opsonised *C. neoformans*

To investigate the role of TLR4 on the phagocytosis of non-opsonised *C. neoformans*, J774A.1 murine macrophages were treated with 0.2 μM TAK-242, an inhibitor of TLR4 signalling, for 1 h before being infected with *C. neoformans* still in the presence of the inhibitor. Surprisingly, TLR4 inhibition resulted in a 1.7-fold increase in the phagocytosis of non-opsonised cryptococci (Fig. 1a). We tested whether genetic loss of *TLR4* would replicate this effect by using immortalised bone marrow derived macrophages (iBMDMs) isolated from wildtype and *Tlr4*−/− C57BL/6 mice. As with the chemical inhibition of TLR4, genetic knockout of *TLR4* led to a pronounced 8-fold increase in the phagocytosis of non-opsonised *C. neoformans* (Fig. 1b, d). To examine the human relevance of this finding, human monocyte derived macrophages (HMDMs) were serum starved for 2 h, pre-treated with 0.2 μM

TAK242 and infected with *C. neoformans*. Similar to the finding in mice cell lines, the loss of TLR4 signalling led to an increase in the non-opsonic phagocytosis of *C. neoformans* (Fig. 1c).

### Increased uptake observed in *Tlr4*−/− macrophages is not a consequence of increased intracellular proliferation

Proinflammatory responses, such as those driven by TLR4, have been shown to restrict the intracellular proliferation of cryptococci[23], hence we considered that the perceived increase in phagocytosis might instead reflect increased proliferation in the absence of TLR4 activity. To test this hypothesis, we conducted live imaging of infected macrophages and quantified the number of internalised fungi at the beginning of the video (T0) and 10 h post infection (T10) to determine the intracellular proliferation rate (IPR). This time-lapse-based IPR assay revealed that neither TLR4 inhibition using TAK-242 (Fig. 1e) or *TLR4* knockout (Fig. 1f) altered the IPR of cryptococci compared to control macrophages. This suggests that the observed intracellular burden of *C. neoformans* is representative of the initial rate of uptake and not due to differences in the subsequent proliferation of the fungi within macrophages.

### Oxidised low-density lipoprotein (ox-LDL) competitively inhibits the phagocytosis of non-opsonised *C. neoformans*

Having shown that the increased intracellular burden of infection following TLR4 inhibition or knockout is a result of elevated phagocytosis and not proliferation, we next sought to identify the plasma membrane receptor responsible for this increase in uptake. Of note, plasma membrane TLRs are not bona fide phagocytic receptors, since they are not directly responsible for the engulfment of whole microorganisms[22]. Consequently, we considered whether TLR4 modulates the availability of one or more phagocytic receptors that then bind non-opsonised *C. neoformans*.

Scavenger receptors, a family of receptors that were initially identified for their role in the uptake of modified host lipoproteins[24], are increasingly being implicated as receptors for a variety of microbes and their ligands[25–27]. Moreover, it has been shown that the expression of several scavenger receptors is upregulated in *Tlr4*−/− mice[28] and that TLR agonists increase the phagocytosis of *Escherichia coli* by inducing the expression of scavenger receptors[29]. We therefore tested whether the loss of TLR4 signalling increases the phagocytosis of non-opsonised *C. neoformans* through the upregulation of scavenger receptors.

Firstly, we treated macrophages with oxidised low-density lipoprotein (ox-LDL), a general scavenger receptor ligand and competitive inhibitor, prior to infection with *C. neoformans*. We found that ox-LDL was able to competitively inhibit the phagocytosis of *C. neoformans* in both wildtype and *Tlr4*−/− macrophages (Fig. 2a, b). When macrophages were infected with 18B7 antibody-opsonised fungi to drive uptake through Fcγ-receptors instead, ox-LDL pre-treatment had no impact on the phagocytosis of cryptococci in both wildtype and *Tlr4*−/− macrophages (Fig. 2c), indicating that the inhibition is specific to non-opsonic uptake. Similar to the finding in a murine cell line, HMDMs pre-treated with ox-LDL also showed reduced phagocytosis of non-opsonic *C. neoformans* (Fig. 2d) though not as significant as in mouse cells, implicating scavenger receptors as broadly relevant receptors involved in the uptake of *C. neoformans*.

### Crosstalk between TLR4 and MSR1 modulate the non-opsonic phagocytosis of *C. neoformans*

Oxidised-LDL is a receptor for a variety of scavenger receptors; therefore, to discern which specific scavenger receptor(s) may be responsible for the phagocytosis of *C. neoformans*, we analysed the surface expression of the scavenger receptors CD36, MAcrophage Receptor with COllagenous structure (MARCO), and

Macrophage Scavenger Receptor 1 (MSR1), also known as SR-A1 or CD204, using flow cytometry. Wildtype and *Tlr4*⁻/⁻ macrophages express high levels of CD36 (Fig. 3a; Supplementary Fig. 1a); however, since the level of CD36 is similar between wildtype and *Tlr4*⁻/⁻ macrophages, it cannot be responsible for the differential level of non-opsonic uptake between these two cell types. Both cell types expressed very little MARCO (Fig. 3b; Supplementary Fig. 1b, x-axis), in line with studies that show that iBMDMs do not express MARCO[30]. Notably, however, MSR1 expression was higher in *Tlr4*⁻/⁻ macrophages compared to wildtype macrophages (Fig. 3c; Supplementary Fig. 1b, y-axis), suggesting that the increased phagocytosis of *C. neoformans* observed in *Tlr4*⁻/⁻ macrophages may be due to their increased expression of MSR1.

Next, to test the direct involvement of individual scavenger receptors in phagocytosis of cryptococci, we infected MPI cells (a non-transformed GM-CSF-dependent murine macrophage cell line[30]) derived from wildtype, *Msr1*⁻/⁻, *Marco*⁻/⁻ or *MSR1/MARCO* double knockout (DKO) C57BL/6 mice with non-opsonised *C. neoformans*.

Whilst uptake by *Marco*⁻/⁻ macrophages was indistinguishable from wildtype cells, macrophages derived from *Msr1*⁻/⁻ mice showed a significant 75% decrease in phagocytosis of non-opsonised cryptococci (Fig. 3d). Double knockout macrophages showed a similar level of reduction in uptake to single knockout *Msr1*⁻/⁻ macrophages, suggesting that the phenotype in double knockout cells is likely due to the loss of MSR1 and not MARCO. Notably, we still observe some non-opsonic uptake in MSR1 and double knockout macrophages, implying a role for one or more additional PRRs in the non-opsonic uptake of cryptococci.

To test whether the TLR4-MSR1 crosstalk observed previously also occurred in these MPI cells, we exposed wildtype MPI cells to the TLR4 inhibitor, TAK-242. Consistent with our previous findings, TR4 inhibition led to an increase in the phagocytosis of non-opsonised cryptococci (Supplementary Fig. 2). Moreover, using confocal microscopy to visualise the distribution of MSR1 on infected *Tlr4*⁻/⁻ macrophages, we observed some accumulation of MSR1 around phagocytic cups containing cryptococci (Supplementary Fig. 3).

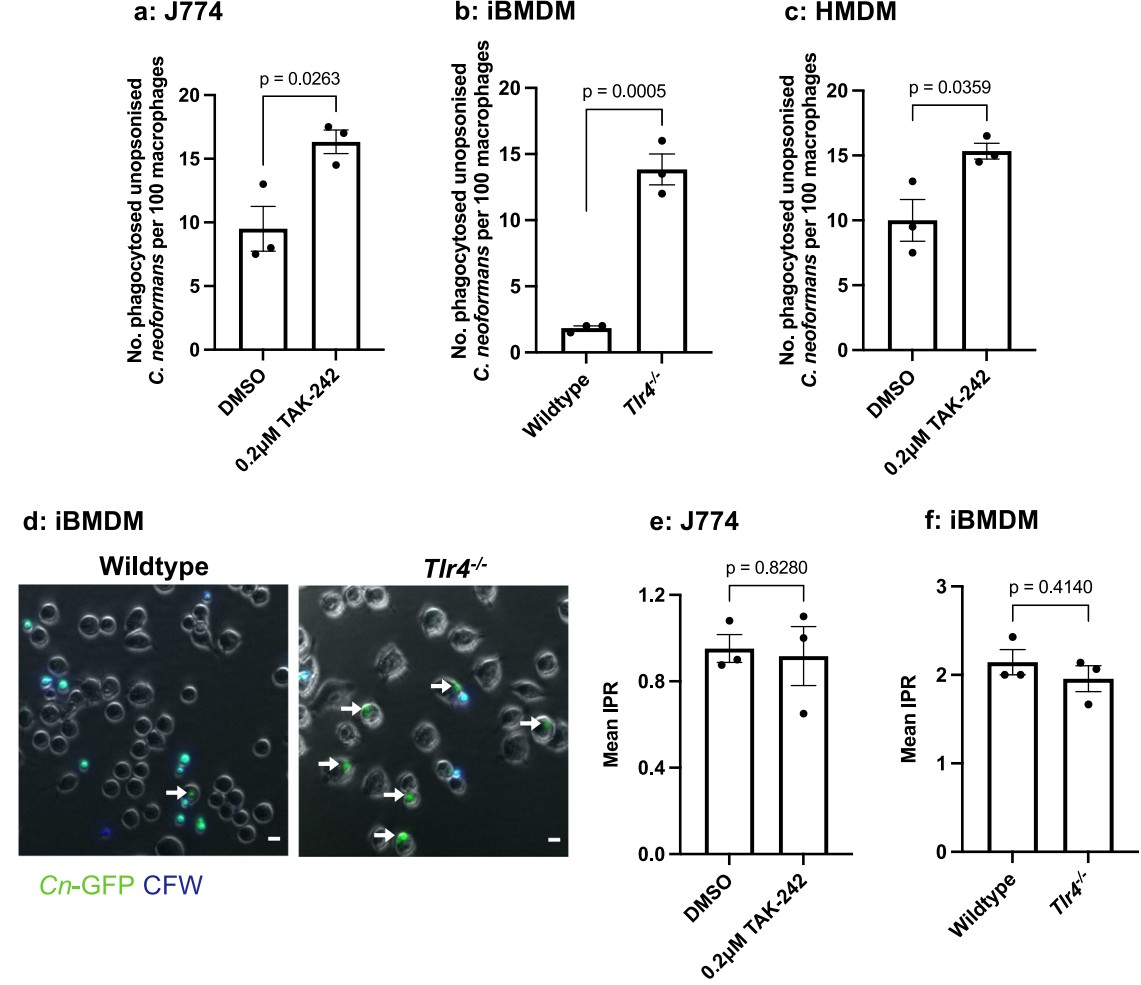

**Fig. 1 | Both chemical inhibition and genetic loss of TLR4 results in an increase in the phagocytosis of *C. neoformans* in both murine and human macrophages.** **a** J774A.1 macrophages or (**c**) human monocyte derived macrophages (HMDMs) were treated with DMSO (control) or 0.2 µM TAK-242, a TLR4 specific inhibitor, for 1 h before infection with non-opsonised *C. neoformans*. **b** Immortalised bone marrow derived macrophages (iBMDM) from wildtype and *Tlr4*⁻/⁻ macrophages were infected with non-opsonised *C. neoformans*. Phagocytosis was quantified as the number of individual internalised cryptococci within 100 macrophages. Figures are representative of at least three independent experiments. HMDM data represents two independent experiments with separate donors. **d** Representative image showing the phagocytosis of GFP-labelled *C. neoformans* (*Cn*-GFP) by wildtype and *Tlr4*⁻/⁻ iBMDM. Calcofluor White (CFW) was used to stain extracellular fungi. White arrows show phagocytosed fungi. Scale bar = 10 µm. The intracellular proliferation (IPR) of *C. neoformans* was measured in (**e**) J774A.1 macrophages and (**f**) wildtype and *Tlr4*⁻/⁻ iBMDMs using timelapse imaging. Images were captured every 5 mins for 18 h. The number of internalised fungi per 100 macrophages at the 'first frame' (T0) and 'last frame' (T10) was quantified and IPR was determined using the equation: IPR = T10/T0. Data is representative of two independent experiments. All data shown as mean ± SEM; *n* = 3 per condition; statistical significance was evaluated using an unpaired two-sided t-test; P-values are shown above each graph. Source data are provided as a Source Data file.

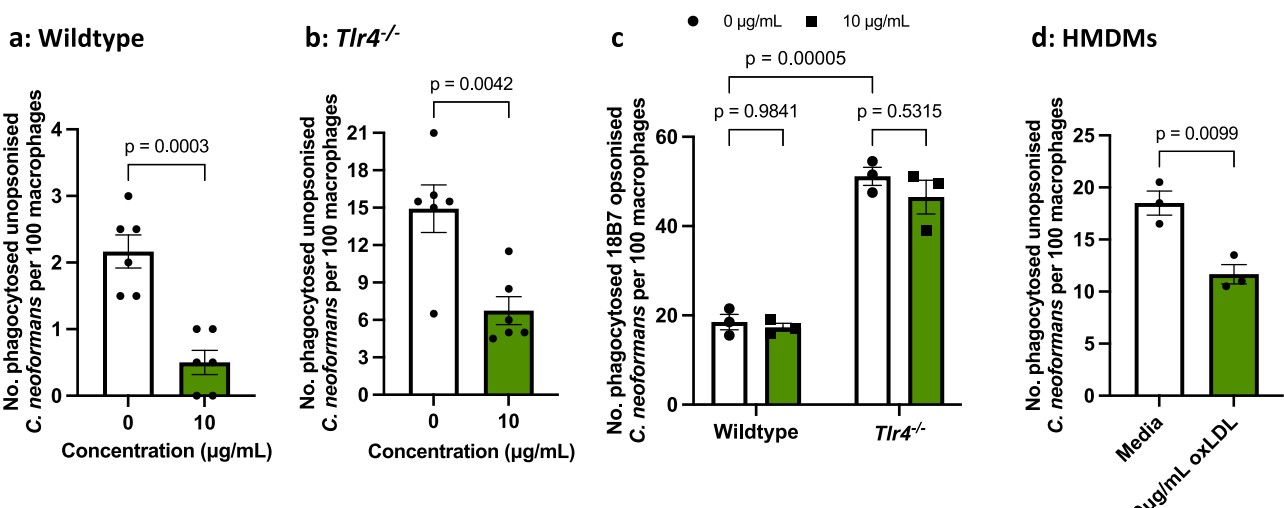

**Fig. 2 | The increased phagocytosis observed in Tlr4⁻/⁻ macrophages is partially driven by scavenger receptors. a** Wildtype iBMDMs (*n* = 6 per condition), **b** *Tlr4*⁻/⁻ iBMDMs (*n* = 6 per condition), and **d** human monocyte derived macrophages (HMDMs) (*n* = 3 per condition) were treated with oxidised low-density lipoprotein (ox-LDL), a general scavenger receptor ligand, for 30 mins prior to infection with non-opsonised *C. neoformans*. iBMDM data is pooled from two independent experiments. HMDM data represents two independent experiments with separate donors. **c** Wildtype and *Tlr4*⁻/⁻ iBMDMs (*n* = 3 per condition) were treated with 10 μg/mL ox-LDL for 30 mins, then infected with *C. neoformans* opsonised with the anti-capsular 18B7 antibody. The number of internalised fungi per 100 macrophages was quantified from fluorescent microscopy images. Data represents two independent experiments. All data is shown as mean ± SEM; statistical significance was evaluated using an unpaired two-sided t-test (**a**, **b**, **d**); or two-way ANOVA followed by Tukey's post-hoc test (**c**). *P*-values are shown above each graph. Source data are provided as a Source Data file.

## TLR3, but not TLR2 or TLR9, is involved in MSR1-mediated phagocytosis without affecting cell surface expression of MSR1

Having identified the existence of crosstalk between TLR4 and MSR1 in the non-opsonic phagocytosis of *C. neoformans*, we then sought to explore the mechanism of MSR1-mediated phagocytosis as well as the mechanism of TLR4-mediated regulation of MSR1 activity. Scavenger receptors lack a clear intracellular signalling domain. It is therefore hypothesised that scavenger receptor-mediated internalisation and proinflammatory cytokine production requires coreceptors and adaptor molecules leading to the formation of multimolecular signalling complexes[26]. Since increased *C. neoformans* uptake in *Tlr4*⁻/⁻ macrophages is likely driven by the increased expression of MSR1 in these cells, we used *Tlr4*⁻/⁻ macrophages as a tool to identify potential proteins involved in MSR1-mediated phagocytosis.

There is evidence of TLR-TLR crosstalk modulating cytokine expression[31]. Consequently, we wondered whether TLR-TLR crosstalk may also influence phagocytosis of cryptococci. Previous work has investigated four different TLRs in the context of *C. neoformans* infection: TLR2, TLR3, TLR4 and TLR9[32–34]. To explore the existence of TLR-TLR crosstalk, we therefore treated wildtype and *Tlr4*⁻/⁻ iBMDMs with inhibitors of TLR2 (CU CPT22), TLR3 (TLR3/dsRNA complex inhibitor), and TLR9 (ODN 2088) prior to infection with *C. neoformans*. Although inhibition of TLR2 and TLR9 had no impact on the non-opsonic uptake of *C. neoformans*, TLR3 inhibition led to a decrease in uptake by wildtype iBMDMs (Fig. 4a). Enhanced phagocytosis seen in *Tlr4*⁻/⁻ macrophages was dampened by TLR3 inhibition, resulting in a 42% decrease in the number of phagocytosed fungi, but not following TLR2 or TLR9 inhibition (Fig. 4b). Interestingly, when macrophages were infected with *C. neoformans* opsonised with the anti-capsular 18B7 antibody, *TLR4*-deficiency still resulted in an increase in uptake, but the effect of TLR3 inhibition was lost in both wildtype and *Tlr4*⁻/⁻ cells (Fig. 4c). Therefore, the role of TLR3 in modulating the phagocytosis of *C. neoformans* is specific to non-opsonic uptake.

To explore the connection between TLR3 and scavenger receptors, *Tlr4*⁻/⁻ iBMDMs were pre-treated with TLR3i and ox-LDL individually and in combination. We first treated macrophages with the effective concentrations of TLR3i (10 μM) and ox-LDL (10 μg/mL) and found that combined treatment did not dampen phagocytosis any more than the individual treatments (Fig. 4d). This suggests that either TLR3 and scavenger receptors act along the same pathway, or that these concentrations of the respective inhibitors are saturating, resulting in no further suppression when both inhibitors are used.

Next, inspired by a study that demonstrated *Msr1*⁺/⁻ or *Tlr4*⁺/⁻ single heterozygote mice showed no impairment in the phagocytosis of *E. coli*, but double heterozygotes were defective in phagocytosis[35], we then tested whether using a lower concentration of ox-LDL and TLR3i individually and in combination would result in synergy. When *Tlr4*⁻/⁻ macrophages were treated with 1 μM TLR3i or 1 μg/mL ox-LDL, there was no difference in the phagocytosis of *C. neoformans* compared to untreated cells (Fig. 4e). However, when treated with 1 μM TLR3i and 1 μg/mL ox-LDL together, there was a decrease in phagocytosis (Fig. 4e). We conclude that both receptors act in synergy along the same pathway since residual 'flux' through the pathway when a low dose of inhibitor is used was further dampened by treatment with both inhibitors.

One possible interpretation of the data presented above would be a model in which loss of TLR4 signalling triggers increased TLR3 signalling, leading to upregulation of scavenger receptors. To test this hypothesis, we measured MSR1 expression in wildtype and *Tlr4*⁻/⁻ iBMDMs after 1 h TLR3 inhibition. However, we found no significant changes in MSR1 expression after both wildtype and *Tlr4*⁻/⁻ macrophages were treated with the TLR3 inhibitor (Fig. 4f), suggesting that the interaction between MSR1 and TLR3 is not at the level of direct TLR3-mediated regulation of MSR1 expression.

## MyD88 and TRIF adaptor proteins modulate non-opsonic phagocytosis of *C. neoformans*

TLR4 and TLR3 signalling require the downstream adaptor molecules MyD88 (used by TLR4) and TRIF (used by both TLR4 and TLR3)[20]. Therefore, to understand the downstream signalling pathway(s) involved, macrophages were exposed to inhibitors of MyD88, TRIF, IKKβ (a kinase downstream of MyD88 that is necessary for NF-κB

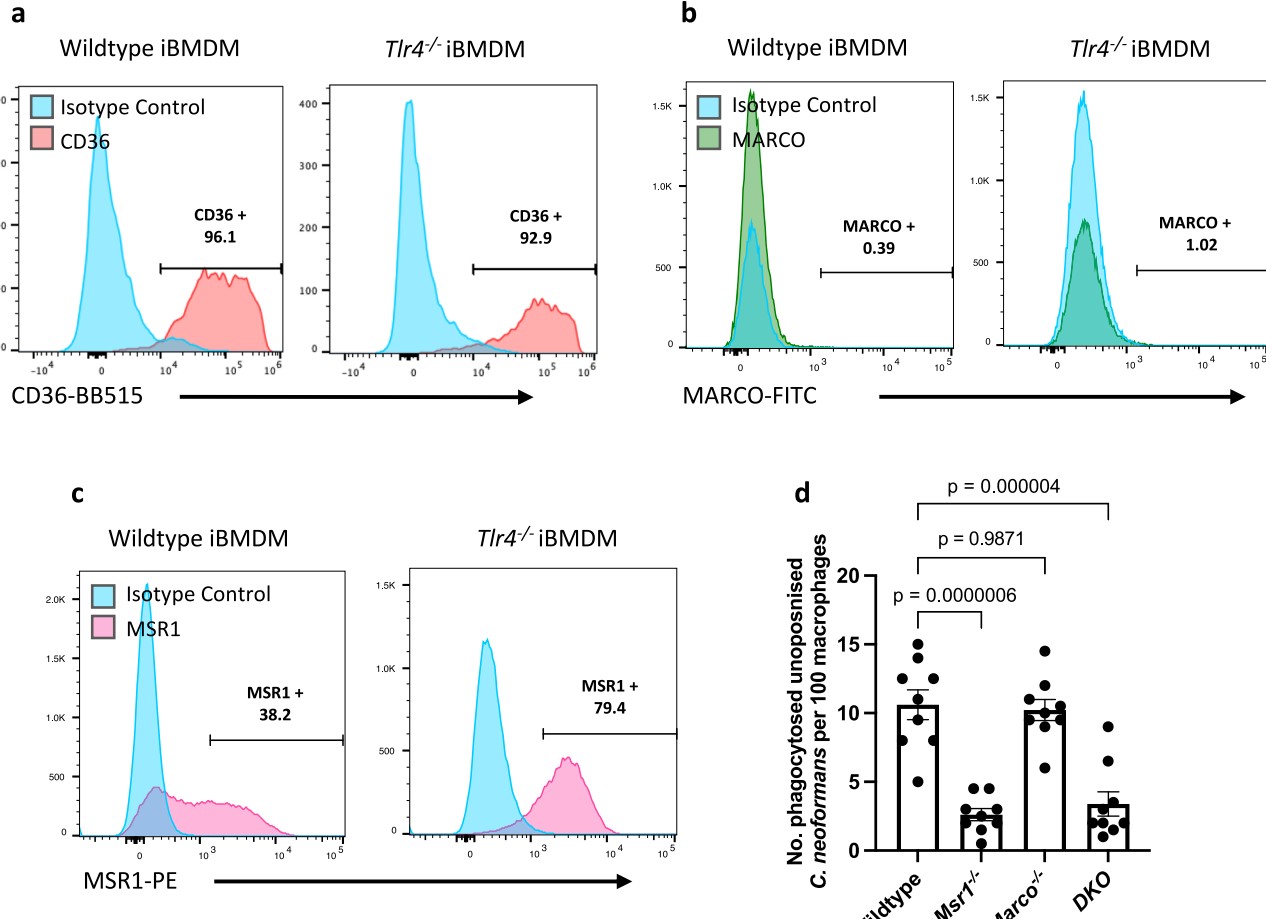

**Fig. 3 | The scavenger receptor, MSR1, mediates the nonopsonic uptake of *C. neoformans*.** Baseline surface expression of **a** CD36 stained with anti-mouse CD36-BB515 antibody; **b** Macrophage Receptor with Collagenous structure (MARCO) stained with anti-mouse MARCO-Fluorescein antibody; and **c** Macrophage Scavenger Receptor 1 (MSR1), also known as CD204, stained with anti-mouse CD204-PE antibody on wildtype and *Tlr4*⁻/⁻ macrophages. Receptor expression was measured using flow cytometry. Data is representative of three independent experiments which gave similar results. Numbers above gates refer to the percentage of CD36-, MARCO- and MSR1-positive cells. **d** MPI cells, a non-transformed GM-CSF-dependent murine macrophage cell line, isolated from wildtype, *Msr1*⁻/⁻, *Marco*⁻/⁻ and *MSR1/MARCO* double knockout (DKO) mice, were infected with non-opsonised *C. neoformans*. The data shown is pooled from three independent experiments (*n* = 9). Data is mean ± SEM; statistical significance was evaluated using a one-way ANOVA; *P*-values are shown above the graph. Source data are provided as a Source Data file.

activation[36]), and TBK1 (a kinase downstream of TRIF that phosphorylates and activates IRF3[37]). In wildtype iBMDMs, treatment with all four inhibitors led to decreased non-opsonic phagocytosis of *C. neoformans* (Fig. 5a). Likewise, all four inhibitors dampened the increased phagocytosis observed in *TLR4*-deficient macrophages (Fig. 5b). In line with the findings from the inhibitor treatments, macrophages derived from *Myd88*⁻/⁻ and *Trif*⁻/⁻ mice were impaired in the phagocytosis of *C. neoformans*, with few or no phagocytosis events being observed in these cells (Fig. 5c).

To ensure that the loss of uptake in *MyD88*- and *TRIF*-deficient macrophages was not caused by an inherent deficiency in phagocytic capacity, we infected iBMDMs with CAF2-dTomato *Candida albicans*[38]. We found that *MyD88*⁻/⁻ and *Trif*⁻/⁻ macrophages had the same level of phagocytosis as wildtype macrophages (Supplementary Fig. 4). Thus, non-TLR dependent pathways such as the Dectin-1 receptor that is recognised as the key PRR involved in the phagocytosis of *C. albicans*[16,17] remain intact in *MyD88*⁻/⁻ and *Trif*⁻/⁻ macrophages. Notably, however, the loss of TLR4 also led to an increase in the phagocytosis of *C. albicans* (Supplementary Fig. 4), suggesting the existence of some shared host response to both fungi. Overall, the phagocytosis of non-opsonised *C. neoformans*, but not *C. albicans*, is dependent on MyD88 and TRIF signalling.

Given that *Myd88*⁻/⁻ and *Trif*⁻/⁻ macrophages showed a near complete loss of phagocytosis, we hypothesised that these cells express very little MSR1. To test this, the surface expression of MSR1 on these macrophages was also measured using flow cytometry. However, as with *Tlr4*⁻/⁻ macrophages, *Myd88*⁻/⁻ and *Trif*⁻/⁻ cells showed increased MSR1 expression compared to wildtype iBMDMs (Fig. 5d). Moreover, the proportion of MSR1 positive cells in *Myd88*⁻/⁻ and *Trif*⁻/⁻ macrophages was similar to that observed in *Tlr4*⁻/⁻ macrophages (20.4% for wildtype, 71.8% for *Tlr4*⁻/⁻, 65.6% for *MyD88*⁻/⁻ and 74.9% for *Trif*⁻/⁻ macrophages (Supplementary Fig. 5)). This implies that increased MSR1 expression alone is not sufficient to drive increased phagocytosis. Either MyD88 and TRIF themselves or some other MyD88- and/or TRIF-dependent molecules may serve as adaptor proteins or coreceptors necessary to drive pathogen engulfment.

**An inhibitor screen identified FcγRII/III, SYK, PI3K, ERK1/2 and p38 involvement in the increased uptake of non-opsonic *C. neoformans* seen in *Tlr4*⁻/⁻ macrophages**

To continue to probe the downstream signalling events that occur during MSR1-mediated uptake, we carried out a small-scale inhibitor screen using *Tlr4*⁻/⁻ macrophages. A previous study investigating a different scavenger receptor, CD36, showed that CD36 forms a

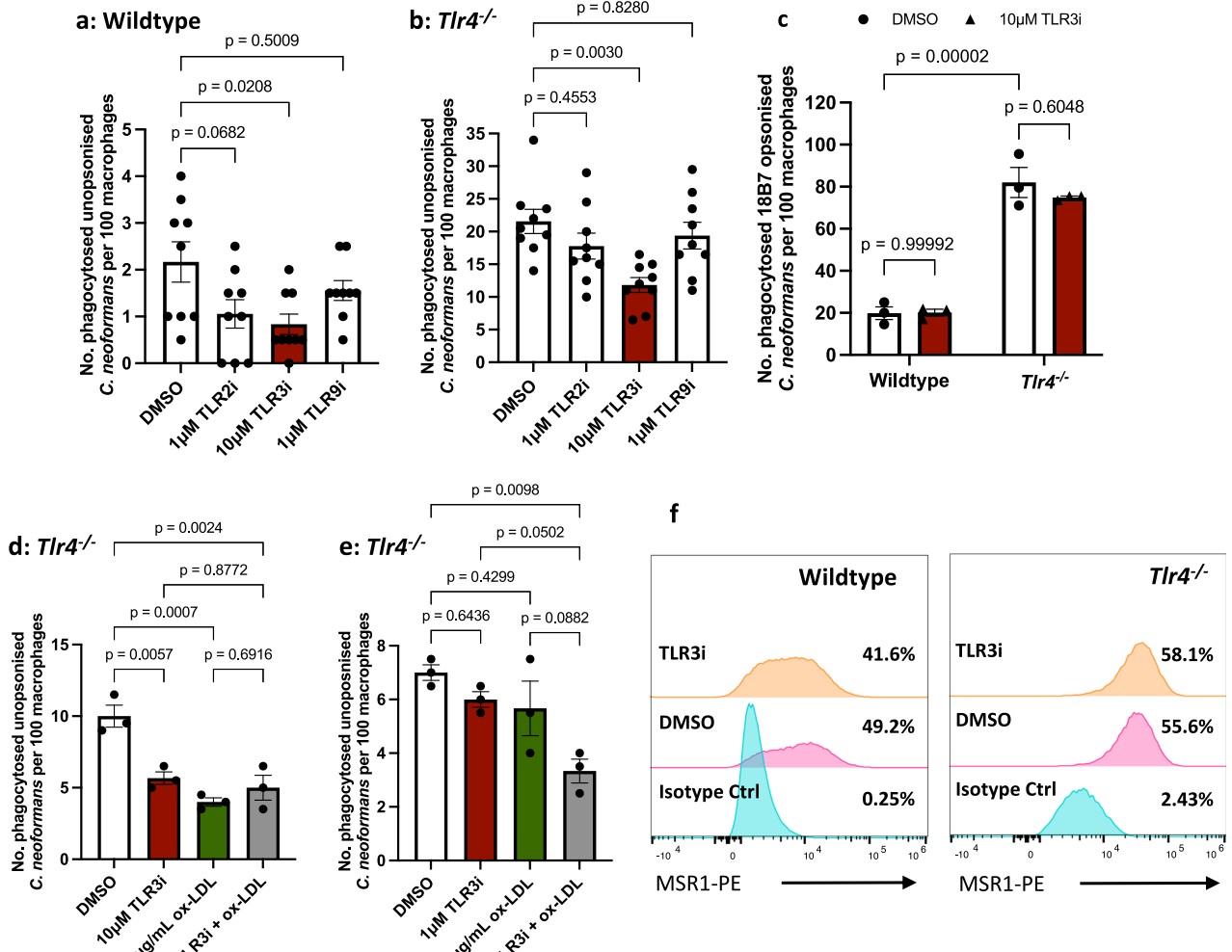

**Fig. 4 | The increased phagocytosis observed in *Tlr4*⁻/⁻ macrophages is dependent on TLR3 signalling. a** Wildtype and **b** *Tlr4*⁻/⁻ iBMDMs were treated with chemical inhibitors of TLR2, TLR3 and TLR9 for 1 h, then infected with non-opsonised *C. neoformans*. Data is pooled from three independent experiments (*n* = 9 per condition). **c** Wildtype and *Tlr4*⁻/⁻ iBMDMs were treated with a TLR3 inhibitor then infected with *C. neoformans* opsonised with the anti-capsular 18B7 antibody (*n* = 3 per condition). **d, e** *Tlr4*⁻/⁻ macrophages were pre-treated with optimal and suboptimal concentrations of TLR3 inhibitor and ox-LDL, individually and in combination (*n* = 3 per condition). Phagocytosis was quantified as the number of internalised cryptococci within 100 macrophages. Data is representative of two independent experiments and is shown as mean ± SEM; statistical significance was evaluated using (**a**, **b**, **d**, **e**) one-way ANOVA or (**c**) two-way ANOVA followed by Tukey's post-hoc test. *P*-values are shown above each graph. **f** Wildtype and *Tlr4*⁻/⁻ iBMDMs were treated with 0.025% DMSO (control) or 10 μM TLR3 inhibitor for 1 h. MSR1 (also known as CD204) expression was measured using anti-mouse CD204-PE monoclonal antibody. PE-labelled rat IgG2a κ was used as an isotype control. Samples were run through a flow cytometer and analysed in the FlowJo software. Percentages refer to the percentage of MSR1-positive cells. Data is representative of two independent experiments. Source data are provided as a Source Data file.

signalling complex involving FcγRs, integrins and tetraspanins[39]. Though FcγRs are known to mediate the phagocytosis of antibody-opsonised *C. neoformans*, inspired by this earlier work we tested whether they might also be involved in aiding non-opsonic phagocytosis through MSR1. Interestingly, blocking FcγRII and FcγRIII using the anti-CD16/CD32 monoclonal antibody led to a decrease in the phagocytosis of non-opsonised cryptococci (Fig. 6a). Thus, it is possible that MSR1 (which has no C-terminal signalling domain of its own) may trigger downstream signals by using the signalling domain of co-clustered FcRs.

Engagement of FcγRs is followed by the phosphorylation of their ITAM motifs[40], followed by recruitment and activation of SYK. Downstream of SYK is phosphoinositide 3-kinase (PI3K), a lipid-modifying enzyme that catalyses the conversion of PtdIns(4,5)P$_2$ into PtdIns(3,4,5)P$_3$ in the plasma membrane[41]. This ultimately leads to a range of biological responses including actin cytoskeleton remodelling to drive phagocytosis[40,41]. Given the putative involvement of FcRs

in MSR1-mediated uptake, we next tested the involvement of these downstream kinases in MSR1-mediated non-opsonic phagocytosis. We found that inhibition of SYK using piceatannol, or of PI3K using wortmannin, led to a dose-dependent decrease in the phagocytosis of cryptococci by macrophages (Fig. 6b). This implies a model where *C. neoformans* binding to MSR1 triggers the formation of a signalling complex with FcγRs. The ITAM domain of FcγRs enables SYK and PI3K activation which then drives actin-remodelling and fungal internalisation (Fig. 7).

It has previously been shown that TLRs induce a phagocytic gene expression program through the mitogen-activated protein kinase p38[29], thus we hypothesised that the increased uptake seen in *Tlr4*⁻/⁻ macrophages may be dependent on MAPK signalling. To test this hypothesis, we pre-treated macrophages with inhibitors of the three classical MAPKs, namely, extracellular signalling kinases (ERK1/2), p38 and c-Jun N-terminal kinase (JNK). ERK1/2 and p38, but not JNK, were involved in non-opsonic phagocytosis (Fig. 6c). MAPKs are also

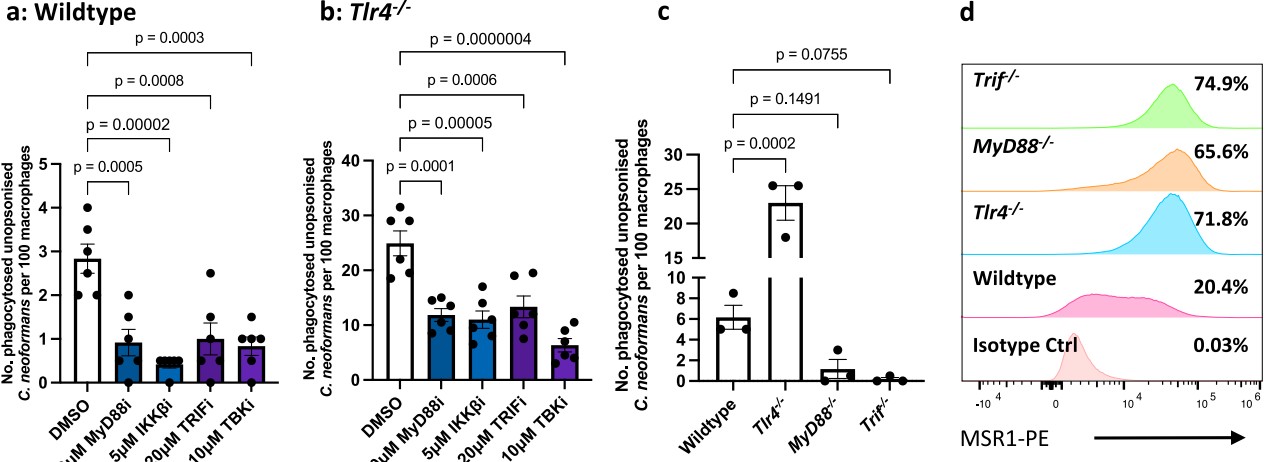

**Fig. 5 | MyD88 and TRIF are required for nonopsonic uptake of *C. neoformans* without affecting MSR1 expression. a** Wildtype and **b** *Tlr4⁻/⁻* macrophages were treated with inhibitors of MyD88, IKKβ (a kinase downstream of MyD88 that is necessary for NF-κB activation), TRIF, and TBK1 (a kinase downstream of TRIF that phosphorylates and activates IRF3) (*n* = 6 per condition). Following pre-treatment with the various inhibitors, cells were infected with non-opsonised *C. neoformans*. Data are pooled from two independent experiments. **c** Immortalised BMDMs from wildtype, *Tlr4⁻/⁻*, *MyD88⁻/⁻* and *Trif⁻/⁻* macrophages were infected with non-opsonised *C. neoformans* (*n* = 3). Data is representative of three independent experiments. Phagocytosis was quantified as the number of internalised cryptococcus within 100 macrophages. Data is mean ± SEM and analysed using one-way ANOVA followed by Tukey's post-hoc test. *P*-values are shown above each graph. **d** Baseline surface expression of Macrophage Scavenger Receptor 1 (MSR1) (also known as CD204) was measured in wildtype, *Tlr4⁻/⁻*, *MyD88⁻/⁻* and *Trif⁻/⁻* macrophages using anti-mouse CD204-PE antibody. PE-labelled rat IgG2a κ was used as an isotype control. Receptor expression was measured using flow cytometry and analysed using the FlowJo software. Data is representative of two independent experiments. Percentages refer to the percentage of MSR1-positive cells. Source data are provided as a Source Data file.

downstream of SYK signalling and have a range of effector functions including proinflammatory cytokine production and cytoskeletal remodelling[42], supporting the proposed model where SYK-recruitment to MSR1 via ITAM-containing coreceptors drives ligand internalisation. The activation of MAPKs is also driven by MyD88 and TRIF signalling[43]. Given the decreased uptake seen following MyD88 and TRIF inhibition (Fig. 5a, b) and in *Myd88⁻/⁻* and *Trif⁻/⁻* macrophages (Fig. 5c), in our model, MyD88 and TRIF would also function as adaptor proteins for MSR1-mediated ligand internalisation (Fig. 7).

Finally, we wondered whether MAPK signalling may modulate baseline MSR1 expression. To test this hypothesis, wildtype and *Tlr4⁻/⁻* macrophages were treated with MAPK inhibitors for 24 h, then MSR1 expression was measured using flow cytometry. However, we found no difference in MSR1 expression following ERK1/2, p38 or JNK inhibition in either wildtype or knockout macrophages (Fig. 6d).

## Discussion

Our findings provide insight into the role of TLR4-MSR1 crosstalk in the phagocytosis of non-opsonised *C. neoformans* by macrophages. We found that the loss of TLR4 signalling unexpectedly increased the phagocytosis of *C. neoformans* by upregulating MSR1 expression, through a yet-to-be-identified mechanism. Using *Msr1⁻/⁻* macrophages we show that MSR1 is a key phagocytic receptor for the uptake of *C. neoformans*, which also explains why other non-opsonic receptors, such as the classical fungal receptor Dectin-1, seem to play a less important role in the host response to *C. neoformans*[5,44]. Notably, since there were still instances of uptake in *Msr1⁻/⁻* macrophages, other non-opsonic receptors must also play a role in the non-opsonic phagocytosis of *C. neoformans*. Such receptors remain to be identified.

Scavenger receptors are phagocytic receptors found on the plasma membrane of various immune cells including macrophages[25]. They were first found to bind modified low-density lipoproteins (LDL), but are now known to recognise a wide range of host and microbial ligands such as apoptotic cells, phospholipids, proteoglycan, LPS, and fungal β-glucans[25–27]. It has previously been reported that TLR4 synergises with MSR1 to promote the phagocytosis of Gram-

negative *E. coli*, while TLR2 synergises with MSR1 in the phagocytosis of Gram-positive *Staphylococcus aureus*[35]. Similarly, MSR1 was involved in the phagocytosis of the Gram-negative bacteria *Neisseria meningitidis*, which is also recognised by TLR4, while modulating TLR4-mediated inflammatory response to *N. meningitidis* infection[45]. Our discovery of TLR/MSR1 crosstalk in the context of a fungal infection therefore suggests that TLR-SR crosstalk as a regulator of phagocytosis and cytokine expression is a general phenomenon of host-pathogen interactions.

Scavenger receptors, including MSR1, have very short cytoplasmic tails with no discernible signalling domains[26], thus it is believed that they act by forming a multimolecular signalling complex. We utilised *Tlr4⁻/⁻* iBMDM as a model to identify potential partners in MSR1-mediated phagocytosis. We first identified a role for TLR3, an endosomal PRR known for its role as a dsRNA receptor[43], in this process. TLR3 inhibition dampened the elevated phagocytosis observed in *Tlr4⁻/⁻* macrophages; however, TLR3 inhibition did not influence MSR1 expression. Since TLR3 is a dsRNA receptor, there is no obvious TLR3 ligand in *C. neoformans*, hence the mechanism driving TLR3 contribution to the modulation of *C. neoformans* uptake by macrophages requires further study.

Despite the decrease in *C. neoformans* phagocytosis observed in *MyD88⁻/⁻* and *Trif⁻/⁻* macrophages, we also found that *MyD88*- and *TRIF*-deficient macrophages had increased expression of MSR1 compared to wildtype macrophages. This suggests that the presence of MSR1 on the cell surface alone is not sufficient to drive phagocytosis. Additionally, it shows that the role of MyD88 and TRIF in the phagocytosis of *C. neoformans* is not due to an impact on MSR1 expression. Instead, they may function as adaptor proteins or activators of some other partner molecule necessary for successful MSR1-mediated pathogen engulfment. The same could be said for TLR3, since treatment with a TLR3 inhibitor had no impact on MSR1 expression even though TLR3 inhibition resulted in decreased phagocytosis.

Further exploration of MSR1-mediated phagocytosis revealed a potential role for FcγRII and III in acting together with MSR1 at the cell surface. This is in line with previous work that found that FcγRs act

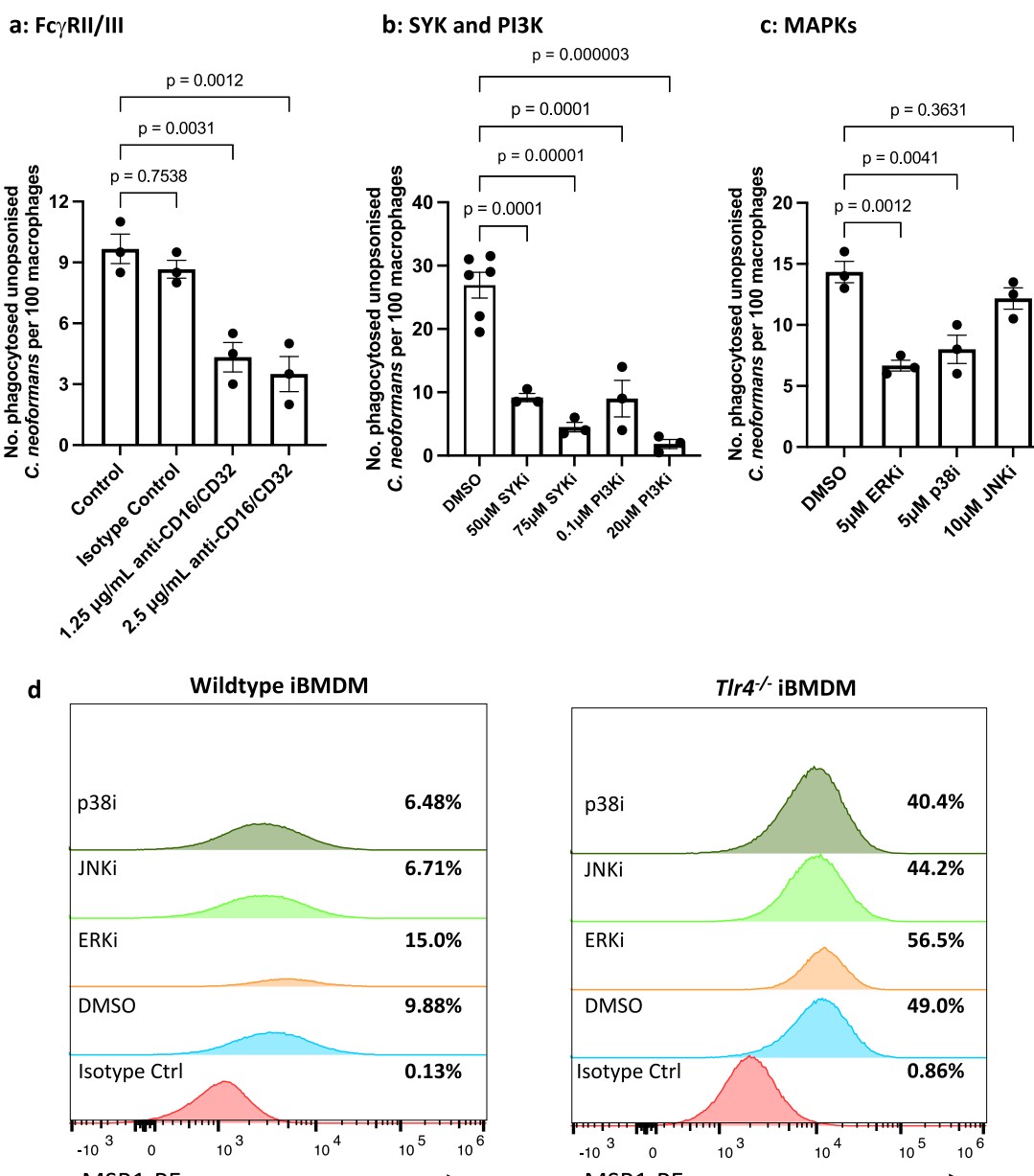

**Fig. 6 | Increased uptake in *Tlr4^-/-* macrophages is dependent on FcγRs, SYK, PI3K, ERK1/2 and p38, but not JNK.** *Tlr4^-/-* macrophages were pre-treated with inhibitors of **a** FcγRs (*n* = 3 per condition), **b** SYK and PI3K (control, *n* = 6; treatment, *n* = 3), and **c** Mitogen Activate Protein Kinases (MAPKs) (*n* = 3 per condition) for 1 h, then infected with non-opsonised *C. neoformans*. The number of internalised fungi per 100 macrophages was quantified from images from a fluorescence microscope. Figures are representative of three independent experiments. Data shown as mean ± SEM; a one-way ANOVA with Tukey's post-hoc test was performed to evaluate statistical significance. *P*-values are shown above each graph. **d** Wildtype and *Tlr4^-/-* iBMDMs were exposed to MAPK inhibitors for 24 h, then cell surface expression of MSR1 was detected with using anti-mouse CD204-PE antibody. PE-labelled rat IgG2a κ was used as an isotype control. Receptor expression was measured using flow cytometry and analysed using the FlowJo software. Data is representative of two independent experiments. Percentages refer to the percentage of MSR1-positive cells. Source data are provided as a Source Data file.

alongside another scavenger receptor, CD36[39]. We also showed that kinases downstream of FcγR, such as SYK and PI3K, and kinases downstream of TLR3, MyD88 and TRIF, such as ERK1/2 and p38, were also involved in non-opsonic uptake. Taken together, we propose a model where TLR4 modulates the surface levels of MSR1. The modulation of cell surface MSR1 may be at the level of TLR4-mediated regulation of MSR1 expression. Alternatively, it may represent a role for TLR4 in redistributing the intracellular reservoir of MSR1, such that TLR4 deficiency drives MSR1 trafficking to the plasma membrane. MSR1 is then responsible for the direct binding and internalisation of *C. neoformans*. The recognition of *C. neoformans* by MSR1 triggers the formation of a signalling complex with FcγRs. The ITAM domain of

FcγRs enables SYK and PI3K activation which then drives actin-remodelling and fungal internalisation. At the same time, TLRs can signal through MyD88 and TRIF to activate ERK1/2 and p38 which will also drive actin remodelling and pathogen uptake.

Although we explore MSR1 signalling via coreceptors, we cannot rule out the possibility of direct MSR1 signalling via its cytoplasmic tail. Human MSR1 has a 50 amino acid-long cytoplasmic tail (55 amino acids long in mice). In silico analysis of the MSR1 protein sequence reveals the presence of a conserved serine residue in humans and mice (serine 27 in humans, and serine 32 in mice), that can be phosphorylated according to UniProt[46], suggestive of a downstream phosphorylation cascade. In fact, it has been shown that a direct association between

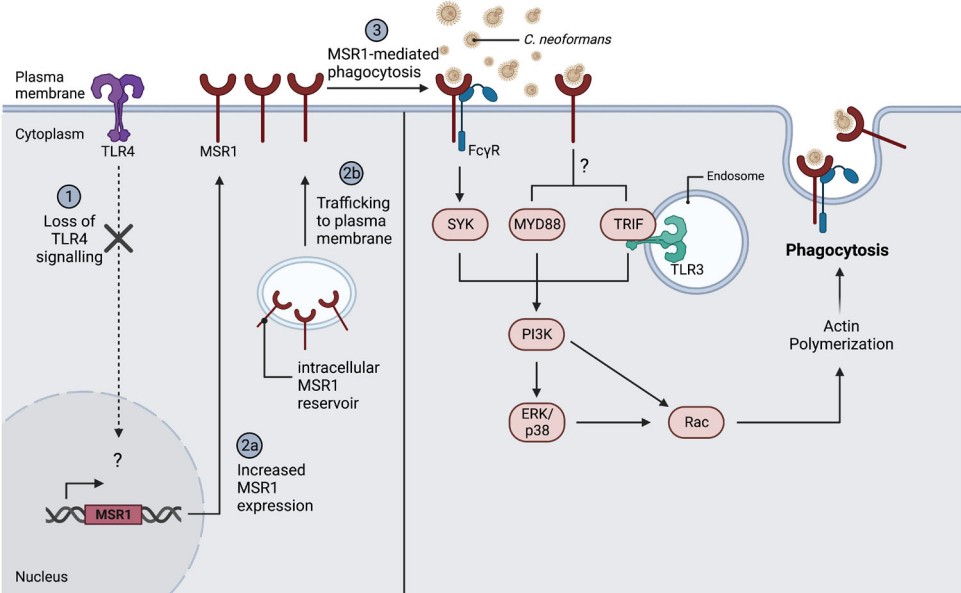

**Fig. 7 | Proposed model of MSR1-meidated non-opsonic phagocytosis.** TLR4 modulates the surface expression of MSR1 through a yet-to-be-identified mechanism, such that loss of TLR4 signalling drives increased surface expression of MSR1. This may be through modulation of MSR1 gene expression or modulation of intracellular MSR1 reservoir trafficking to the plasma membrane. MSR1 is then responsible for the direct binding and internalisation of *C. neoformans*. MSR1-mediated uptake may rely on the formation of a signalling complex with FcγRII/III. The ITAM domain of FcγRs enables SYK and PI3K activation which then activates Rac GTPases that drive actin polymerisation and phagocytosis. At the same time, TLR3, MyD88 and TRIF may also serve as coreceptors or adaptor proteins leading to the activation of ERK1/2 and p38 which will also drive actin remodelling and pathogen uptake. Figure created with BioRender.com.

MSR1 and Mer receptor tyrosine kinase is needed to drive the uptake of apoptotic cells by macrophages[47]. Similarly, Lyn tyrosine kinase has been co-immunoprecipitated with MSR1 during uptake of acetylated LDL[48], supporting the possibility of direct MSR1-mediated signalling.

The mechanism of TLR4-mediated regulation of MSR1 expression remains elusive. We measured MSR1 expression following exposure to inhibitors of TLR3 and MAPKs and in *MyD88*[−/−] and *Trif*[−/−] macrophages. None of these signalling pathways had any effect on MSR1 expression even though they played a role in phagocytosis. Therefore, the mechanism of TLR4/MSR1 crosstalk to modulate phagocytosis may be independent of the mechanism controlling MSR1 expression. Notably ERK1/2 has been shown to modulate LPS-induced MSR1 expression[49], but not CD36 expression[50]. Moreover, JAK-STAT signalling modulated MSR1 and CD36 expression following LPS stimulation. We did not see any role for ERK1/2 in modulating MSR1 expression in our study, likely because we did not use LPS stimulation in our experimental design. In the future, it will be interesting to investigate the mechanism of MSR1 expression at baseline, without LPS stimulation, perhaps through an omics-based approach.

Others have investigated the role of TLR4 during the host response to *Cryptococcus* infection; however, these studies have revealed contradictory results[9,32,51,52]. An in vitro study found that the stimulation of microglial cells isolated from the brain of wildtype mice with the TLR4 agonist, lipopolysaccharide (LPS), resulted in increased phagocytosis and killing of *C. neoformans* in a MyD88-dependent manner[53]. Interestingly, in vivo studies using *TLR4*-deficient mice have found that the receptor is dispensable during host response to infection[32,33,51]. MyD88 is a key adaptor molecule downstream of all TLRs except TLR3. Mice deficient in *MyD88* consistently show that this adaptor molecule plays a major role in anti-*Cryptococcus* immune response[33,51], thereby implicating the upstream TLRs in host response. However, to date, the precise role of individual TLRs, including TLR4, during cryptococcal infection is poorly understood. Our data suggest that one possible explanation for these previous conflicting results is varying level of MSR1 expression, which was unaccounted for in these studies. Here we show that the knockout of *TLR4* increased MSR1 expression; however, others have shown that LPS-mediated stimulation of TLR4 was also capable of increasing the expression of scavenger receptors leading to increased uptake[29,49]. Despite expecting TLR4-deficiency and pre-treatment with a TLR4 agonist to have opposing effects, our data imply that any perturbation of TLR4 signalling affects scavenger receptor expression which could impact macrophage response to infection.

An in vivo study using *Msr1*[−/−] mice found that knockout mice had reduced lung fungal burden and decreased expression of T-helper 2 (Th2) cytokines, which is an immune polarisation state that promotes fungal growth and dissemination[13]. Thus, the authors concluded that MSR1 is normally hijacked by *C. neoformans* to promote its pathogenesis. If this is the case, the increased expression of MSR1 that we observe in *Tlr4*[−/−] macrophages could correlate with poor disease outcomes. In support of this idea is the finding that *C. neoformans* clinical isolates that are more readily phagocytosed showed increased brain fungal burden, reduced mice survival and polarization towards the non-protective Th2 response[54]. Similarly, clinical isolates with low phagocytic indexes were associated with poor fungal clearance (even with antifungal treatment) in the cerebrospinal fluid[55]. Meanwhile, isolates with high phagocytic indexes were associated with increased mortality[55,56]. Therefore, both very high and very low phagocytosis are predictors of poor disease outcome, implying the existence of a 'Goldilocks' level of uptake. Ultimately, strategies to manipulate non-opsonic uptake of cryptococci (perhaps with blocking inhibitors or (ant)agonists of the uptake pathway) may prove a useful therapeutic approach in achieving that 'Goldilocks' level in patients, although given that uptake is not driven by MSR1 alone, we must consider the contribution of other non-opsonic PRRs and possible crosstalk between receptors. In addition, it may also be relevant to explore the effect of MSR1 polymorphisms on susceptibility to cryptococcal meningitis.

In summary, here we present the significance of TLR4/MSR1 crosstalk in the phagocytosis of non-opsonised *C. neoformans*, identify MSR1 as a critical receptor for the non-opsonic phagocytosis of

*C. neoformans* and propose a mechanism whereupon ligand binding, MSR1 interacts with coreceptors such as FcγRs to trigger downstream signalling.

## Methods

### Ethics statement

Leucocyte cones used for this study were obtained with ethical approval from the Science, Technology, Engineering and Mathematics Ethical Review Committee at the University of Birmingham (approval reference ERN15_0804c). Informed consent was obtained prior to blood donation. All samples were fully anonymised and destroyed after experimentation.

### Tissue culture and macrophage cell lines

The J774A.1 cell line [ECACC] was cultured in T-75 flasks [Fisher Scientific] in Dulbecco's Modified Eagle medium, low glucose (DMEM) [Sigma-Aldrich], containing 10% live fetal bovine serum (FBS) [Sigma-Aldrich], 2 mM L-glutamine [Sigma-Aldrich], and 1% Penicillin and Streptomycin solution [Sigma-Aldrich] at 37 °C and 5% $CO_2$. During phagocytosis assays, J774A.1 macrophages were seeded at a density of $1 \times 10^5$ cells per well of a 24-well plate [Greiner Bio-One] and incubated overnight at 37 °C and 5% $CO_2$.

Immortalised bone marrow-derived macrophages were originally isolated from C57BL/6 wildtype, *Tlr4*[-/-], *MyD88*[-/-] and *Trif*[-/-] single knock-out mice and immortalised via transformation with retroviruses expressing Raf and Myc, two well-known proto-oncogenes[57–59]. Immortalised BMDMs were cultured in DMEM, low glucose [Sigma-Aldrich] supplemented with 10% heat-inactivated FBS [Sigma-Aldrich], 2 mM L-glutamine [Sigma-Aldrich], and 1% Penicillin and Streptomycin solution [Sigma-Aldrich] at 37 °C and 5% $CO_2$. During phagocytosis assays, iBMDMs were seeded at a density of $3 \times 10^5$ cells per well of a 24-well plate [Greiner Bio-One] twenty-four hours prior to infection. The cells were then incubated overnight at 37 °C and 5% $CO_2$.

Max Plank Institute (MPI) cells are a non-transformed, granulocyte-macrophage colony-stimulating factor (GM-CSF)-dependent murine macrophage cell line that is functionally similar to alveolar macrophages[30,60]. In this study, MPI cells from wildtype, *Msr1*[-/-], macrophage receptor with collagenous structure knockout (*Marco*[-/-]) and *MSR1/MARCO* double knockout (DKO) C57BL/6 mice were utilised. Cells were cultured in Roswell Park Memorial Institute (RPMI) 1640 medium [ThermoFisher] supplemented with 10% heat-inactivated FBS [Sigma-Aldrich], 2 mM L-glutamine [Sigma-Aldrich], and 1% Penicillin and Streptomycin solution [Sigma-Aldrich] at 37 °C and 5% $CO_2$. Each flask was further supplemented with 1% vol/vol GM-CSF-conditioned RPMI media prepared using the X-63-GMCSF cell line. When being used in phagocytosis assays, MPI cells were seeded at a density of $2 \times 10^5$ cells per well of a 24-well plate [Greiner Bio-One] with 1% vol/vol GM-CSF. The cells were then incubated overnight at 37 °C and 5% $CO_2$. All cell lines used in this research are available commercially or through the authors.

### Isolation of monocytes and macrophage differentiation

Leucocyte cones, from healthy donors, were obtained from the National Health Service Blood Transfusion Service (NHSBT) and promptly processed within 4 hours. Peripheral blood mononuclear cells (PBMCs) were isolated from the cones using a standard density gradient centrifugation method. Following a DPBS wash, the blood was mixed with an equal volume of DPBS (containing 2% FBS) and carefully layered onto Lymphoprep (StemCell). Centrifugation was carried out at $1100 \times g$ for 20 minutes without applying brakes. The resulting white buffy layer was collected and further washed with DPBS by centrifuging at $300 \times g$ for 10 minutes. Red blood cells (RBC) were removed by using an RBC lysis solution (BioLegend) for 10 minutes at room temperature. CD14+ monocytes were then isolated from the PBMCs using immunomagnetic positive selection (Miltenyi Biotec). The isolated

human monocytes were seeded at a density of $0.5 \times 10^6$ cells per well in tissue culture-treated 24-well plates and cultured in complete RPMI 1640 medium supplemented with 10% Human Serum and 20 ng/mL human GM-CSF (PeproTech) for macrophage differentiation. Cells were incubated for 6 days, with medium replacement on day 3.

### Phagocytosis assay

Phagocytosis assays were performed to measure the uptake of *Cryptococcus* by macrophages under various conditions. Twenty-four hours before the start of the phagocytosis assay, an overnight culture of *Cryptococcus neoformans* var. *grubii* KN99α strain, that had previously been biolistically transformed to express green fluorescent protein (GFP)[61], was set up by picking a fungal colony from YPD agar plates (50 g/L YPD broth powder [Sigma-Aldrich], 2% Agar [MP Biomedical]) and resuspending in 3 mL liquid YPD broth (50 g/L YPD broth powder [Sigma-Aldrich]). The culture was then incubated at 25 °C overnight under constant rotation (20 rpm).

On the day of the assay, macrophages were activated using 150 ng/mL phorbol 12-myristate 13-acetate (PMA) [Sigma-Aldrich] for 1 h at 37 °C. PMA stimulation was performed in media containing heat-inactivated serum (iBMDMs and MPI cells) or in serum-free media (J774A.1) to eliminate the contribution of complement proteins during phagocytosis. When using human monocyte-derived macrophages (HMDMs), M0 macrophages were serum starved for 2 h. Where applicable, macrophages were then treated with the desired concentration of soluble inhibitors of PRRs (Supplementary Data 1) and incubated at 37 °C for 1 h. Meanwhile, pre-treatment with the general scavenger receptor ligand, oxidised low-density lipoprotein (ox-LDL), occurred for 30 mins. The concentration used for each molecule is indicated in Supplementary Data 1 and in the corresponding results.

To prepare *C. neoformans* for infection, the overnight *C. neoformans* culture was washed two times in 1X PBS and centrifuged at 6500 rpm for 2.5 mins. To infect macrophages with non-opsonised *C. neoformans*, after the final wash, the *C. neoformans* pellet was resuspended in 1 mL PBS, counted using a hemacytometer, and fungi incubated with macrophages at a multiplicity of infection (MOI) of 10:1. The infection was allowed to take place for 2 h at 37 °C and 5% $CO_2$. Infection occurred in the presence of soluble inhibitors.

In some instances, macrophages were infected with antibody-opsonised *C. neoformans*. To opsonise the fungi, $1 \times 10^6$ yeast cells in 100 μL PBS were opsonised for 1 h using 10 μg/mL anti-capsular 18B7 antibody (a kind gift from Arturo Casadevall, Johns Hopkins University, Baltimore, USA). After 2 h infection, macrophages were washed 4 times with PBS to remove as much extracellular *C. neoformans* as possible.

### Fluorescent microscopy imaging

Having washed off extracellular cryptococci, the number of phagocytosed fungi was quantified using images from a fluorescent microscope. To distinguish between phagocytosed and extracellular *C. neoformans*, wells were treated with 10 μg/mL calcofluor white (CFW) [Sigma-Aldrich], a fluorochrome that recognises cellulose and chitin in cell walls of fungi, parasite and plants[62], for 10 mins at 37 °C. Next, fluorescent microscopy images were acquired using the Zeiss Axio Observer [Zeiss Microscopy] fitted with the ORCA-Flash4.0 C11440 camera [Hamamatsu] at 20X magnification. The phase contrast objective, EGFP channel and CFW channel were used. Image acquisition was performed using the ZEN 3.1 Blue software [Zeiss Microscopy] and the resulting images were analysed using the Fiji image processing software [ImageJ].

To quantify the number of phagocytosed cryptococci, the total number of ingested *C. neoformans* was counted in 200 macrophages, then the values were applied to the following equation: ((number of phagocytosed *C. neoformans*/number of macrophages) * 100). Therefore, the result of the phagocytosis assay is presented as the number of internalised fungi per 100 macrophages.

## Live imaging

To assess the intracellular proliferation rate (IPR) of *C. neoformans* within macrophages, infected macrophages were captured at a regular interval over an extended period. Live-cell imaging was performed by running the phagocytosis assay as usual, then after washing off extracellular cryptococcus, the corresponding media for the macrophage cell line was added back into the well before imaging. Live imaging occurred using the Zeiss Axio Observer at 20X magnification and images were acquired every 5 mins for 18 h at 37 °C and 5% $CO_2$.

The resulting videos were analysed using Fiji [ImageJ] and IPR was determined by quantifying the total number of internalised fungi in 200 macrophages at the 'first frame' (time point 0 (T0)) and 'last frame' (T10). Then, the number of phagocytosed fungi at T10 was divided by the number of phagocytosed fungi at T0 to give the IPR (IPR = T10/T0).

## Immunofluorescent imaging

Immunofluorescence was used to investigate receptor localisation on macrophages. Firstly, 13 mm cover slips were placed onto 24-well plates prior to seeding with the desired number of macrophages. After overnight incubation, macrophages were used in a standard phagocytosis assay. Prior to staining, macrophages were fixed with 4% paraformaldehyde for 10 mins at room temperature and permeabilised with 0.1% Triton X-100 diluted in PBS for 10 mins at room temperature. To prevent non-specific binding, cells were incubated with 1% bovine serum albumin (BSA) diluted in 1xPBS for 30 mins at room temperature. To stain for MSR1 localisation, rabbit anti-mouse MSR1 (E4H1C) (1:100) [Cell Signalling Technology; Cat#: 91119; Clone E4H1C] was used as the primary antibody. Cells were incubated with the primary antibody for 1 h at room temperature. After washing three times with PBS, macrophages were incubated with Alexa Fluor 594 conjugated anti-rabbit IgG F(ab')$_2$ fragment secondary antibody (1:500) [Cell Signalling Technology; Cat#: 8889 S] for 1 h at room temperature and in the dark. Coverslips were mounted on 5 μL VECTASHIELD HardSet antifade mounting medium with DAPI [Vector Laboratories]. Images were acquired using the Zeiss LSM880 Confocal with Airyscan2, laser lines 405, 488, 561 and 640 nm, and at 63X oil magnification. Image acquisition was performed using the ZEN Black 3.0 software [Zeiss Microscopy] and the resulting images were analysed using the Fiji image processing software [ImageJ].

## Flow cytometry

Flow cytometry was used to measure the surface expression of scavenger receptors on macrophages. Prior to staining, macrophages were incubated with 2.5 μg/mL rat anti-mouse CD16/CD32 Fc block [BD Biosciences; Cat#: 553142; Clone 2.4G2] diluted in FACS buffer (1XPBS without $Mg^{2+}$ and $Ca^{2+}$ supplemented with 2% heat-inactivated FBS and 2 mM EDTA). After Fc blocking, the desired concentration of fluorochrome-conjugated antibodies diluted in FACS buffer was added into each tube still in the presence of the Fc block mixture. The following fluorochrome-conjugated antibodies were used: 0.5 μg/mL anti-mouse CD45-PerCP-Cyanine5.5 [ThermoFisher; Cat#: 45-0451-82; Clone 30-F11], 0.25 μg/100 μL anti-mouse CD204(MSR1)-PE [Fisher Scientific; Cat#: 12-204-682; Clone M204PA], 0.25 μg/100 μL anti-mouse CD36-BB515 [BD Biosciences; Cat#: 565933; Clone CRF D-2712], and 10 μL/100 μL anti-mouse MARCO-Fluorescein [Biotechne; Cat#: FAB2956F; Clone 579511]. Fluorescent minus one (FMO) controls were included to aid in setting gating boundaries. Isotype controls were used to test for non-specific binding. The following isotype control antibodies were used: 0.25 μg/100 μL PE rat IgG2a, κ isotype control [Fisher Scientific; Cat#: 15248769; Clone eBR2a], 0.25 μg/100 μL BB515 Mouse IgA, κ isotype control [BD Biosciences; Cat#: 565095; Clone M18-254], and 10 μL/100 μL Rat IgG$_1$ Fluorescein isotype control [Biotechne; Cat#: IC005F; Clone IC0057]. Finally, samples stained with only one fluorophore were used as compensation controls. After staining, samples were resuspended in FACS buffer for single staining and unstained controls and FACS buffer with 0.2 μg/mL DAPI [ThermoFisher], a live dead stain, for all other samples.

Stained samples were run on the Attune NxT flow cytometer [ThermoFisher] and acquired using the Attune NxT software [ThermoFisher]. The resulting data was analysed using the FlowJo v10.8.1 software for MacOS [BD Life Sciences]. Before determining the proportion of macrophages positive for a particular fluorochrome, a gating strategy was employed to achieve the sequential exclusion of debris and doublets (Supplementary Fig. 6). Anti-CD45-PerCP-Cyanine5.5 was used to identify total leucocytes, and DAPI was used to exclude dead cells.

## Statistics

GraphPad Prism version 9.5.0 for MacOS (GraphPad Software, San Diego, CA) was used to generate graphical representations of numerical data. Inferential statistical tests were performed using Prism. The data sets were assumed to be normally distributed based on the results of a Shapiro-Wilk test for normality. Consequently, to compare the means between treatments, the following parametric tests were performed: unpaired two-sided t-test, one-way ANOVA, and two-way ANOVA. ANOVA tests were followed up with Tukey's post-hoc test. Variation between treatments was considered statistically significant if *p*-value < 0.05.

## Reporting summary

Further information on research design is available in the Nature Portfolio Reporting Summary linked to this article.

## Data availability

Source data are provided with this paper.

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

## Acknowledgements

We thank Maria Makarova for assistance with confocal microscopy, Michael G. Tomlinson for his in-silico analysis of MSR1 and contribution to the discussion, Sean D. Kelly for technical help regarding the establishment of MPI cell lines and Sarah Dimeloe and Nancy Gudgeon for helpful advice on the culture of PBMCs. The *MSR1* knockout and *MSR1/MARCO* double knockout lines were established with the support of NC3Rs Grant NC/V001019/1. Work in C.E.B. lab is supported by grant BB/V000276/1. This work was partly supported by a Royal Society project grant (RGS\R2\202260) to S.M lab. C.U.O is supported by a Ph.D. studentship from the Darwin Trust of Edinburgh. A.L.W. was supported by Ph.D. funding from the Wellcome Trust 'MIDAS' doctoral training program. R.C.M. gratefully acknowledges support from the BBSRC and European Research Council Consolidator Award.

## Author contributions

R.C.M. conceived the project. A.L.W and G.D. generated preliminary data that influenced the conception of the project. C.U.O. and R.C.M. generated hypotheses and designed the experiments. C.U.O performed all the experiments, analysed the data and wrote the manuscript. G.D. helped perform the flow cytometry experiments. S.L.R. and E.M.F. isolated monocytes from human leucocyte cones. C.E.B provided Tlr4$^{-/-}$, MyD88$^{-/-}$, Trif$^{-/-}$ immortalised BMDMs. S.M., S.G. and G.F. provided Msr1$^{-/-}$, Marco$^{-/-}$ and DKO MPI cells. O.D.C provided transfected cell lines that contributed to hypothesis generation. S.M. and S.G. provided critical discussion of the experimental results. All authors contributed to the review of the manuscript. R.C.M. acquired funding.

## Competing interests

The authors declare no competing interests.
