## [Peer Review File · Nature Communications]

Toll-Like Receptor 4 and Macrophage Scavenger Receptor 1 crosstalk regulates phagocytosis of a fungal pathogenREVIEWER COMMENTS

Reviewer #1 (Remarks to the Author):

The authors demonstrate (unexpectedly) that TLR4-deficiency in bone marrow-derived macrophages coincides with increased expression of the scavenger receptor (MSR1/SRA) and increased phagocytosis of nonopsonized *C. neoformans*. MSR1 is therefore a bona fide phagocytic scavenger receptor. They also show that the increase in non-opsonic phagocytosis was driven by crosstalk between TLR4 and TLR3 in a MyD88- in TRIF-dependent manner. The data supports MSR1 as a major pattern recognition receptor for phagocytosing nonopsonized *C. neoformans*, providing an explanation for the minimal involvement of Dectin-1 in host response to *C. neoformans* as reported by others. The authors are the first to report that TLR/MSR1 crosstalk extends from bacterial to fungal pathogen phagocytosis, acting as a general phenomenon of host pathogen interactions.

The data is sound and of high quality. The results are novel and the paper is well written and presented. However, the phagocytosis they observe is only partially driven by scavenger receptors (Fig. 4) so another mechanism must operate and be taken into account when thinking about therapeutic options designed to target nonopsonic phagocytosis of *C. neoformans* at its primary site of infection, i.e. the lung environment.

It is also unclear what contribution nonopsonic phagocytosis of *C. neoformans* by tissue-resident macrophages plays, as opposed to uptake of shed fungal components like cell wall and/or capsule sugar to initiate the adaptive immune response. Furthermore, macrophages recruited from the blood are clearly seen surrounding cryptococcal cell masses (cryptococcomas) to form granulomas in mouse lung histology sections. Hence phagocytosis per se may not be needed to control infection or drive an adaptive immune response. Could the authors comment on this?

The authors findings also contradict those of others who used TLR4 knockout mice to demonstrate that a functional TLR4 receptor does not contribute greatly to resistance to *C. neoformans* infection [PMID: 15322035 (46, 47)], and that MSR1-/- knockout mice had reduced lung fungal burden (consistent with blocking rather than increasing MSR1 expression being a better form of therapy). In the discussion, the authors summarize from the literature that both very high and very low phagocytosis of *C. neoformans* are predictors of poor disease outcome in mouse models, implying the existence of a 'Goldilocks' level of uptake, and that more understanding of the clinical outcomes associated with increased compared to decreased phagocytosis of *C. neoformans* will point towards the appropriate way to manipulate MSR1 as a potential therapeutic approach. However, it seems that manipulating MSR1 or TLR4 levels as a potential therapeutic approach to achieve this 'Goldilocks' level of uptake, would be difficult to achieve. Hence it is unclear in an infection scenario whether tinkering with the expression levels of TLR4 and/or MSR1 would be a good therapeutic option.

Comments and questions

1. Inclusion of a model summarizing the findings would be highly beneficial to the reader, given the complexity of TLR4 signaling mediated by MyD88 and TRIF, leading to pathogen engulfment, including the involvement of NF- κ B, MAPKs, IRF3, Type I interferons, Rap GTPase/Rac GTPase TLRs and phagocytic integrins/receptors. The model should show the proposed mechanism by which TLR4-TLR3 crosstalk regulates expression and/or activity of MSR1. This section of the discussion was difficult to follow. Their flow data implies that increased MSR1 expression alone in Tlr4 negative cells, is not sufficient to drive increased phagocytosis, and that either MyD88 and TRIF or MyD88- and/or TRIF-dependent molecules may serve as adaptor proteins or coreceptors necessary to drive pathogen engulfment. This should also be shown in the model.

2. The number of replicates in each figure legend needs more explanation. In Fig 1-7, data is pooled from 2 technical replicates. Does this mean 1 biological replicate with 2 technical repeats? Is there enough statistical power? In Figure 8, the data is pooled from 3 technical repeats, each performed in triplicate. Do the authors mean 3 biological replicates?

3. It is not clear why non transformed MPI cells were used instead of immortalized BMDMs in Figure 8? Were BMDMs also used and was the same result obtained?
4. TLR3 cross talk study shows approx. 50% reduction in non-opsonic Cryptococci uptake (Fig 3A). Hence another mechanism for opsonic uptake must exist. Do the authors have an idea of what this mechanism might be?
5. Fig 4D and E: Why are the results additive for TLR3 inhibitor and oxLDL combination only for the low (1 μ m) conc not the high (10 μ m) conc.
6. Fig 4E low dose inhibitors: Please clarify what is meant by "acting in synergy in the same pathway" Doesn't the fact that the decrease is bigger with the combination of inhibitors as opposed to with individual inhibitors, suggest that separate pathways are involved?
7. Line 215-216: Can the data showing the proportion of MSR1 positive cells in MyD88^{-/-} and Trif^{-/-} macrophages being similar to that observed in Tlr4^{-/-} macrophages (20.4% for wildtype, 71.8% for Tlr4^{-/-}, 65.6% for MyD88^{-/-} and 74.9% for Trif^{-/-} macrophages), be shown in the supplement?
8. Presumably Fig 3C is measuring uptake of non-opsonised crypto. It should be clear in the results and the figure legend
9. Line 149 phagocytosis as wildtype macrophages (Error! Reference source not found.).
10. Line 152 "the loss of TLR4 also led to an increase in the phagocytosis of *C. albicans* (Error! Reference source 153 not found.)

Reviewer #2 (Remarks to the Author):

The manuscript by Onyishi and colleagues investigates the role of several important pattern recognition receptors for the phagocytosis of *Cryptococcus neoformans*. The authors report the surprising observation that TLR4 deficiency increases the uptake of *Cryptococcus*, process that is mediated by TLR3 and MSR1. Through complementary experiments, the authors demonstrate that MSR1 is a major recognition receptor for *Cryptococcus*, and important for the phagocytosis of this microorganism. The experiments are in general well-performed and the manuscript is clearly written.

Comments:

1. In Figure 3, the TLR3 inhibitor seems to inhibit phagocytosis, but the results are not significant due to the lack of power (only 3 samples/mice). I think it would be good to expand the number, but to test reproducibility and to obtain significant differences.
2. In general, the number of points compared in the various experiments tends to be low (often n=3), this should be improved.
3. The relevance of the study would be greatly increased if some of the main findings would be also validated in human cells.
4. Are the MSR1 knock-out mice more susceptible to cryptococcosis?
5. There are a number of known human polymorphisms in TLR4 gene that influence the function of the receptor. Can the authors speculate whether these polymorphisms could be important for susceptibility to cryptococcosis.

Response to Reviewers' Comments

We include below the unedited comments from both reviewers, together with our response (at the appropriate place in the text), shown in *red*.

Reviewer #1 (Remarks to the Author):

The authors demonstrate (unexpectedly) that TLR4-deficiency in bone marrow-derived macrophages coincides with increased expression of the scavenger receptor (MSR1/SRA) and increased phagocytosis of nonopsonized *C. neoformans*. MSR1 is therefore a bona fide phagocytic scavenger receptor. They also show that the increase in non-opsonic phagocytosis was driven by crosstalk between TLR4 and TLR3 in a MyD88- in TRIF-dependent manner. The data supports MSR1 as a major pattern recognition receptor for phagocytosing nonopsonized *C. neoformans*, providing an explanation for the minimal involvement of Dectin-1 in host response to *C. neoformans* as reported by others. The authors are the first to report that TLR/MSR1 crosstalk extends from bacterial to fungal pathogen phagocytosis, acting as a general phenomenon of host pathogen interactions.

The data is sound and of high quality. The results are novel and the paper is well written and presented. However, the phagocytosis they observe is only partially driven by scavenger receptors (Fig. 4) so another mechanism must operate and be taken into account when thinking about therapeutic options designed to target nonopsonic phagocytosis of *C. neoformans* at its primary site of infection, i.e. the lung environment.

We have now amended the manuscript to make clear that we still observe phagocytosis even in MSR1 deficient macrophages, pointing to the involvement of other yet to be identified phagocytic receptor(s) (Lines 181-183; 441-444; 566-567).

It is also unclear what contribution nonopsonic phagocytosis of *C. neoformans* by tissue-resident macrophages plays, as opposed to uptake of shed fungal components like cell wall and/or capsule sugar to initiate the adaptive immune response. Furthermore, macrophages recruited from the blood are clearly seen surrounding cryptococcal cell masses (cryptococcomas) to form granulomas in mouse lung histology sections. Hence phagocytosis per se may not be needed to control infection or drive an adaptive immune response. Could the authors comment on this?

The role of phagocytosis in the response to cryptococcal infection is double-sided. Whilst efficient phagocytosis and killing helps remove cryptococcal cells, phagocytic uptake in the absence of efficient killing is believed to help drive fungal dissemination. Consequently, studies have found a correlation between the rate of uptake and mortality in murine models (PMID: 30520685; PMID: 24743149). Our work reveals a hitherto unknown importance for MSR1 and TLR4 signalling in regulating phagocytic uptake of non-opsonised cryptococci, which is likely to be particularly important at an early stage of lung infection. Determining whether MSR1-driven uptake is helpful to the host (in aiding clearance) or the pathogen (in aiding dissemination) is a fascinating question that will require future investigation using murine models and, ultimately, clinical investigation.

The authors findings also contradict those of others who used TLR4 knockout mice to demonstrate that a functional TLR4 receptor does not contribute greatly to resistance to *C. neoformans* infection [PMID: 15322035 (46, 47)], and that MSR1^{-/-} knockout mice had reduced lung fungal burden (consistent with blocking rather than increasing MSR1 expression being a better form of therapy). In the discussion, the authors summarize from

the literature that both very high and very low phagocytosis of *C. neoformans* are predictors of poor disease outcome in mouse models, implying the existence of a 'Goldilocks' level of uptake, and that more understanding of the clinical outcomes associated with increased compared to decreased phagocytosis of *C. neoformans* will point towards the appropriate way to manipulate MSR1 as a potential therapeutic approach. However, it seems that manipulating MSR1 or TLR4 levels as a potential therapeutic approach to achieve this 'Goldilocks' level of uptake, would be difficult to achieve. Hence it is unclear in an infection scenario whether tinkering with the expression levels of TLR4 and/or MSR1 would be a good therapeutic option.

Previous work on TLR involvement in cryptococcal infection is somewhat contradictory. The paper cited above exploited a TLR4 knockout in the murine C3H/HeJ background, but an independent TLR4 knockout in the C57Bl/6 background (which is the same background as the cells used in our manuscript) showed no role for TLR4 in defense against cryptococcal infection. A similar scenario exists for TLR2, which has been implicated in cryptococcal defense by some authors but not by others (e.g. <https://doi.org/10.1111/j.1574-695X.2006.00078.x> versus <https://doi.org/10.1002/eji.200425799>). We suggest that one reason for this is precisely the 'Goldilocks' hypothesis that the reviewer highlights – presumably a role (or lack thereof) for TLRs may depend heavily on aspects such as the infectious dose, route of infection and so forth.

We agree entirely with this reviewer that 'tinkering with expression' of either TLR4 or MSR1 may not represent a sensible therapeutic strategy. However, one might imagine alternative approaches, such as blocking antibodies to prevent MSR1-driven uptake (if uptake proves to be deleterious due to increased dissemination). Needless to say, such approaches are still many years away from delivery, but nonetheless we hope that our work represents a first step towards such developments. We have now amended our discussion section to make these points clearer in the manuscript (Lines 560-569).

Comments and questions

1. Inclusion of a model summarizing the findings would be highly beneficial to the reader, given the complexity of TLR4 signaling mediated by MyD88 and TRIF, leading to pathogen engulfment, including the involvement of NF- κ B, MAPKs, IRF3, Type I interferons, Rap GTPase/Rac GTPase TLRs and phagocytic integrins/receptors. The model should show the proposed mechanism by which TLR4-TLR3 crosstalk regulates expression and/or activity of MSR1. This section of the discussion was difficult to follow. Their flow data implies that increased MSR1 expression alone in Tlr4 negative cells, is not sufficient to drive increased phagocytosis, and that either MyD88 and TRIF or MyD88- and/or TRIF-dependent molecules may serve as adaptor proteins or coreceptors necessary to drive pathogen engulfment. This should also be shown in the model.

We have now added a model to the paper, as suggested (Figure 7).

2. The number of replicates in each figure legend needs more explanation. In Fig 1-7, data is pooled from 2 technical replicates. Does this mean 1 biological replicate with 2 technical repeats? Is there enough statistical power? In Figure 8, the data is pooled from 3 technical repeats, each performed in triplicate. Do the authors mean 3 biological replicates?

We agree that the use of 'technical repeats' and/or 'biological repeats' was rather confusing. Instead of using 'technical repeats' and/or 'biological repeats', we now simply state the number of independent experiments performed for each data set.

3. It is not clear why non transformed MPI cells were used instead of immortalized BMDMs in Figure 8? Were BMDMs also used and was the same result obtained?

Non transformed MPI cells were used in this context because they are the only scavenger receptor knockout cells currently available. In addition, MARCO is not expressed by iBMDMs, so using MPI cells allowed us to explore the involvement of this receptor in uptake. We were careful to check that MPI cells and BMDMs behave similarly for the purposes of our experiments, by cross-validating their response to the TLR4-inhibitor TAK242, and now include these data as Supplementary Figure 2.

4. TLR3 cross talk study shows approx. 50% reduction in non-opsonic Cryptococci uptake (Fig 3A). Hence another mechanism for opsonic uptake must exist. Do the authors have an idea of what this mechanism might be?

Though we mainly focus on the role of MSR1 in nonopsonic uptake, other scavenger receptors may be involved. For instance, our flow data identified high levels of CD36 in immortalized BMDMs from both wildtype and Tlr4-/- mice (Figure 3a), so CD36 could also be involved in uptake. However, since the level of CD36 expression is similar between WT and Tlr4-/- cells, it cannot be responsible for the differential level of non-opsonic uptake between these two cell types, which is why we did not study it further in the context of this manuscript. We now clarify this in the revised manuscript [Lines 161-164].

5. Fig 4D and E: Why are the results additive for TLR3 inhibitor and oxLDL combination only for the low (1 μ m) conc not the high (10 μ m) conc.

We tested various concentrations of TLR3 inhibitor and oxLDL and found that 10 μ M and 10 μ g/mL were enough to give the maximal effect. So the absence of an additive/synergistic effect at high concentrations may reflect saturation of each receptor individually and/or antagonism between the different inhibitors (e.g. via steric hindrance).

6. Fig 4E low dose inhibitors: Please clarify what is meant by “acting in synergy in the same pathway” Doesn't the fact that the decrease is bigger with the combination of inhibitors as opposed to with individual inhibitors, suggest that separate pathways are involved?

We are grateful to the reviewer for pointing out that this sentence was confusing. We have now rephrased this part of the manuscript, along with the associated figure, accordingly. The logic behind this experiment is that a) at high dose, there is no synergy between the two inhibitors. Therefore if MSR1 signalling is fully 'switched off' through OxLDL, there is no further suppression by the inactivation of TLR3. Conversely, if MSR1 is only partially inhibited by low dose OxLDL, then the residual 'flux' through this pathway can be further dampened by (partially) inhibiting TLR3. We interpret this as meaning both receptors interact on the same pathway.

7. Line 215-216: Can the data showing the proportion of MSR1 positive cells in MyD88-/- and Trif-/- macrophages being similar to that observed in Tlr4-/- macrophages (20.4% for wildtype, 71.8% for Tlr4-/-, 65.6% for MyD88-/- and 74.9% for Trif-/- macrophages), be shown in the supplement?

We have now included this data as Supplementary Figure 5.

8. Presumably Fig 3C is measuring uptake of non-opsonised crypto. It should be clear in the results and the figure legend

In the figure legend of Fig 3C (now Fig 5a-b) we have now stated that macrophages were infected with non-opsonised cryptococcus. We have also made it clear within the results chapter that non-opsonized cryptococcus was used (Line 266).

9.Line 149 phagocytosis as wildtype macrophages (Error! Reference source not found.).

10.Line 152 “the loss of TLR4 also led to an increase in the phagocytosis of C. albicans (Error! Reference source 153 not found.)

Our apologies – these two cross-referencing errors were not visible in our submitted manuscript file, but we have now removed all automatic cross-referencing within the text to avoid them recurring.

Reviewer #2 (Remarks to the Author):

The manuscript by Onyishi and colleagues investigates the role of several important pattern recognition receptors for the phagocytosis of *Cryptococcus neoformans*. The authors report the surprising observation that TLR4 deficiency increases the uptake of *Cryptococcus*, process that is mediated by TLR3 and MSR1. Through complementary experiments, the authors demonstrate that MSR1 is a major recognition receptor for *Cryptococcus*, and important for the phagocytosis of this microorganism. The experiments are in general well-performed and the manuscript is clearly written.

Comments:

1. In Figure 3, the TLR3 inhibitor seems to inhibit phagocytosis, but the results are not significant due to the lack of power (only 3 samples/mice). I think it would be good to expand the number, bot to test reproducibility and to obtain significant differences.

We have performed additional repeats of the experiment shown in Figure 3A (now Fig4a-b) and confirmed the reproducibility and significance of the finding.

2. In general, the number of points compared in the various experiments tends to be low (often n=3), this should be improved.

We have now clarified what we mean by the number of technical replicates and independent biological experiments in relation to the comment raised by reviewer 1. All of our data is derived from two or more independent experiments, typically with three or more technical replicates within each, so the real ‘n’ number is actually much greater than 3, as is hopefully now clear in the manuscript.

3. The relevance of the study would be greatly increased if some of the main findings would be also validated in human cells.

We agree and so have now undertaken two additional experiments to address this, using human monocyte-derived macrophages (HMDMs). Firstly, we show that HMDMs pretreated with TAK242 (TLR4 inhibitor) exhibit the same increase in phagocytosis that is seen in murine cells (Figure 1c). Secondly, we show that oxidized LDL competitively inhibit the nonopsonic uptake of *C. neoformans* by these human cells, as it does in murine cells, supporting the relevance of scavenger receptors in the uptake of cryptococcus by both species (Fig 2d).

4. Are the MRS1 knock-out mice more susceptible to cryptococcosis?

We have not directly tested the susceptibility of MSR1 knockout mice to cryptococcosis. However, another group has shown that MSR1^{-/-} mice had decreased lung fungal burden and a skew towards a protective Th1/M1 immune response; suggesting that MSR1 is hijacked by the fungi to promote disease progression (PMID: 23733871). Understanding the interplay between non-opsonic uptake and resistance/susceptibility to cryptococcosis in vivo will be a key question for future research.

5. There are a number of known human polymorphisms in TLR4 gene that influence the function of the receptor. Can the authors speculate whether these polymorphisms could be important for susceptibility to cryptococcosis.

We have previously published a review on human genetic polymorphisms associated with risk of cryptococcosis (PMID: 34716931). SNPs in TLRs, including one in TLR4 (rs1927907), have been linked to susceptibility to cryptococcal meningitis in non-HIV patients (PMID: 30366814), although the mechanism by which these SNPs influence TLR activity is yet to be studied. The Toll-Like Receptor 4 Asp299Gly (rs4986790) and Thr399Ile (rs4986791) polymorphisms are one of the most well studied cosegregating polymorphisms of TLR4. The polymorphisms result in LPS hypo-responsiveness and have been linked to susceptibility to infection and cancers. Some have found that the polymorphisms lead to reduced TLR4 expression on cell surface (PMID: 20212095) and that Asp299Gly polymorphism confers a minor structural change on TLR4 which may affect ligand binding, surface expression and stability (PMID: 23055527). The effect of these polymorphisms on susceptibility to cryptococcosis is yet to be investigated; however, it is possible that SNPs that impact TLR4 expression would in turn influence the expression and activity of nonopsonic receptors leading to changes in host response to infection. It could also be interesting to investigate the effect of MSR1 polymorphisms on cryptococcosis risk. Some commonly studied SNPs of MSR1 include rs3747531 (P275A missense variant) which is associated with prostate cancer (PMID: 17768178) and COPD (PMID: 20081102), and rs13306550, a splice donor site variant associated with risk of COPD (PMID: 29731913) and hypertension (PMID: 26716027). Their role in risk of infectious diseases is yet to be investigated.

REVIEWERS' COMMENTS

Reviewer #1 (Remarks to the Author):

The authors have substantially revised their manuscript and addressed all of the queries. The results are novel, the paper is well written and presented and the data sound and of high quality, supporting MSR1 as a major pattern recognition receptor for phagocytosing nonopsonized *C. neoformans*, predominantly in the context of a mouse model. Addition of the human data as requested by reviewer 2 is also appreciated to provide more clinical relevance, although the increase in the non-opsonic phagocytosis of *C. neoformans* associated with loss of TLR4 signalling is not as significant as in the mouse (iBMDM) cells. Similarly, the decrease in *C. neoformans* phagocytosis in human cells with OxLDL treatment (to block scavenger receptors) is not as significant as in the mouse cells.

The inclusion of a model to summarize the findings is appreciated and suggests increased expression of MSR1 upon loss of TLR4 signalling. Although the flow cytometry results clearly show more MSR1 on the surface, did the authors determine whether this is due to enhanced expression of the scavenger receptor upon TLR4 suppression using qPCR? If there is no change in MSR1 expression, would this infer that intracellular scavenger receptor reservoirs are recruited to the periphery upon TLR4 suppression? Hence the model would require slight modification.

Reviewer #2 (Remarks to the Author):

The authors responded appropriately to my comments.

Point-by-point response to the reviewers' comments

Reviewer #1 (Remarks to the Author):

The authors have substantially revised their manuscript and addressed all of the queries. The results are novel, the paper is well written and presented and the data sound and of high quality, supporting MSR1 as a major pattern recognition receptor for phagocytosing nonopsonized C. neoformans, predominantly in the context of a mouse model.

Addition of the human data as requested by reviewer 2 is also appreciated to provide more clinical relevance, although the increase in the non-opsonic phagocytosis of C. neoformans associated with loss of TLR4 signalling is not as significant as in the mouse (iBMDM) cells. Similarly, the decrease in C. neoformans phagocytosis in human cells with OxLDL treatment (to block scavenger receptors) is not as significant as in the mouse cells.

The inclusion of a model to summarize the findings is appreciated and suggests increased expression of MSR1 upon loss of TLR4 signalling. Although the flow cytometry results clearly show more MSR1 on the surface, did the authors determine whether this is due to enhanced expression of the scavenger receptor upon TLR4 suppression using qPCR? If there is no change in MSR1 expression, would this infer that intracellular scavenger receptor reservoirs are recruited to the periphery upon TLR4 suppression? Hence the model would require slight modification.

We are grateful for these insightful comments, which have improved the quality of our manuscript. We have now included a statement highlighting the fact that the impact of scavenger receptor mediated uptake appears stronger in murine cells than human ones [Lines 145].

We agree with the reviewer's suggestion that the increased surface display of MSR1 may reflect redistribution from intracellular pools, and so have updated the model diagram to include this alternative possibility. A statement has been included in Lines 357 to 360 and the figure has been modified accordingly.

Reviewer #2 (Remarks to the Author):

The authors responded appropriately to my comments.

Thank you for your insightful review of our manuscript. We appreciate all your feedback and suggestions which has helped strengthen our paper.